# Lessons Learned from Introducing Last Aid Courses at a University Hospital in Germany

**DOI:** 10.3390/healthcare9070906

**Published:** 2021-07-16

**Authors:** Evelyn Mueller, Georg Bollig, Gerhild Becker, Christopher Boehlke

**Affiliations:** 1Medical Center, Department of Palliative Medicine, Faculty of Medicine, University of Freiburg, 79106 Freiburg, Germany; evelyn.mueller@uniklinik-freiburg.de (E.M.); gerhild.becker@uniklinik-freiburg.de (G.B.); 2Palliative Care Team, Medical Department Sønderborg/Tønder, South Jutland Hospital, 6400 Sønderborg, Denmark; georg.bollig@rsyd.dk; 3Palliative Care Research Group, Medical Research Unit, Institute of Regional Health Research, University of Southern Denmark, 6200 Aabenraa, Denmark; 4Last Aid International, 24837 Schleswig, Germany

**Keywords:** Last Aid course, palliative care, hospital staff, education, survey

## Abstract

In recent years, so called “Last Aid courses”, concerning end-of-life care for people dying, have successfully been established in community settings in several European countries, Australia, and South-America. To date, they have not been evaluated in hospital settings, where educational needs (concerning care of the dying) are especially high, and may differ from the general population. To evaluate if Last Aid courses are feasible in hospital settings, and if informational needs of hospital staff are met by the curriculum, we introduced Last Aid courses at a university hospital. Five courses were offered; participants of courses 1 and 2 completed surveys with open-ended questions; the answers were used to develop the evaluation questionnaire employed in courses 3–5. In these three courses, 55 of the 56 participants completed an evaluation survey to explore their learning goals and obtain feedback. Courses were fully booked; participants were heterogeneous with regard to their professional background. The most prevalent learning goals were “preparation for emotional aspects in care of dying” (65.5% ratings “very important”), “preparation for medical/care aspects in care of dying” (60.0%), and “knowledge of supportive services and facilities” (54.5%). Overall, the evaluation showed that Last Aid courses were more suitable to educate non-medical hospital staff about care of the dying. Medical staff, in contrast to non-medical staff, more often requested courses with an extended curriculum in order to meet their learning goals. Last Aid courses were well accepted and helped to reduce information deficits on care of the dying in a heterogeneous population of hospital staff.

## 1. Introduction

Estimations of individuals in need of general palliative care at the end-of-life exceed 60% [1,2]; numbers will increase in the decades to come due to expected demographic changes [3]. While the medical needs of seriously ill and dying people must be met by professionals, many needs (e.g., practical and emotional support) can be supported by citizens and compassionate communities [4]. Citizens of a compassionate community care for the old, sick, and dying people in their midst; “death” is integrated as a part of life in society. However, the public is often not aware of the needs of dying people and their relatives, or people are inadequately educated or prepared to support them [5].

The Last Aid course curriculum was created by an international working group to educate citizens about the care of the dying and palliative care; it was successfully implemented in community settings in several countries [6,7,8]. Furthermore, a special Last Aid course curriculum was developed for children and teenagers, and is well accepted by them [9].

Hospital staff (medical and non-medical) face death and dying in various ways; thus the need for information on palliative care is possibly even higher than for the general population. At the same time, professional courses on palliative care are not common in many medical disciplines, and are not intended for non-medical professions. We therefore implemented Last Aid courses with only four teaching hours (45 min each) in a hospital setting, and asked if the courses were suited for the needs of the staff. We investigated the participants’ reasons for course attendance, their professional and private burdens (concerning death and dying), general feedback on the course, and suggestions for improvements.

## 2. Materials and Methods

We implemented Last Aid courses in a tertiary university hospital in Germany in 2018. Each course was open to 20 participants. The courses were announced for non-medical staff on the hospital’s intranet platform. However, courses were open to all employees (including medical staff) of the hospital and no participation fees applied. The curriculum was comprised of four sessions, 45-min, named, “dying as a normal part of life”, “planning ahead”, “relieving suffering”, and “final goodbyes”. Teaching methods included lectures, group discussions, and practical exercises (for details, see [6]). Overall, five courses took place. Each course was fully booked and 17–20 persons participated. The interest in the courses was high—another five courses could have been held without need for additional announcements, but courses were paused due to time constraints of the teachers (two physicians, three nurses). Courses were taught in teams of two (physician and nurse).

### 2.1. Pre-Survey and Item Development

In the first two courses, participants were asked to complete a pre-survey of mainly open questions, so researchers could collect information for item development for the survey in courses 3–5. The pre-survey was completed by 33 participants (n = 4 medical staff; n = 29 non-medical staff; n = 27 female, n = 2 male, n = 2 no answer; age: m = 49.6 (sd = 10.2)). The pre-survey resulted in written descriptions on various topics, e.g., their goals and concerns (see Appendix A), as well as their burdens, due to death and dying. The answers were categorized by clustering similar descriptions and formulating subcategory labels by C.B. and E.M. For goals and expectations, five subcategories were identified. The subcategory descriptions formed the basis for the item wording of the “goals and concerns before attending the course” in the survey, of courses 3–5 (see Appendix B; item 8). For “burden due to death and dying”, only generalized items on burdens in professional work and private life were included (see Appendix B; items 4 and 5). A range of other items was developed based on information from literature and the pre-survey (see Appendix B; not all results reported).

All newly developed items were pre-tested regarding face validity, suitability of Likert scales for questions, and comprehensibility through three cognitive interviews (physician, nurse, social worker), and modified by a team of three experts in the field (palliative medicine, psychology, and psychometrics). As the sample size in the pre-survey and cognitive interviews was limited, we included open-ended questions in the questionnaire in courses 3–5, ensuring participants could add new aspects, if they were not reflected in the pre-survey.

### 2.2. Evaluation Survey

In courses 3–5, the actual evaluation survey was conducted, the results of which are reported here. Participants were informed that the evaluations—aimed at further development of the course and data collection, analysis, and reporting—were anonymous. They were asked to complete the survey immediately after the course or send the form back via mail. The questionnaire is included in Appendix B. The survey comprised: (a) sociodemographic and occupational characteristics of participants; (b) items for evaluation of course contents, extent of new information, recommendation, and comprehensibility (drawn from standard evaluation questionnaires); (c) the specially developed items on goals and concerns, as well as burden due to death and dying. The items employed Likert-scales as answer options and open-ended questions. Additionally, participants could include comments and suggestions in an open-ended question.

### 2.3. Data Analysis

For analysis, medical expertise levels were obtained by classification of reported professions into four levels: 0 non-medical professions; 1 therapeutic and medical assistance staff; 2 nurses and midwifes; and 3 physicians. Furthermore, burden due to death and dying was asked both for work and for private context. Within a summarized category of “overall burden”, any person reporting medium, high, or very high burden in at least one of the contexts was classified as “burdened by death and dying”.

Missing data were not imputed. Explorative data analysis included calculation of descriptive statistics as well as correlations (Kendall Tau-b); the Friedman test was employed to test for differences in ratings between the four modules; due to the explorative approach of the analysis, alpha level was not adjusted and was 5 % (two-tailed) for all tests. SPSS 24 was used for statistical analysis.

## 3. Results

Courses 3–5 were fully booked (n = 60), and 56 staff members attended the course; 55 of the 56 (98.2%) participants completed the evaluation survey. Characteristics of the sample are reported in Table 1. The sample was heterogeneous with regard to age, profession, and extent of patient contact. A total of 90.9% (n = 50) of the participants were female. The participants covered the full range of professions in a hospital, from hairdressers, or laboratory and administrative staff (with no medical expertise) to nurses and physicians. The majority of participants (67.3% (n = 37)) was non-medical staff. A total of 36.4% (n = 20) of the participants reported care or support for dying relatives in the private context.

### 3.1. Motivation to Participate in Last Aid Courses and Burden by Death and Dying

To understand the motivation of participants, we asked the participants of courses 1 and 2 for their reasons to attend the course. For examples of open-ended answers in the pre-survey categorization, see Appendix A. We deducted categories from the answers, formulated items on that basis, and asked the participants in the main survey how important these goals were (Figure 1). “Preparation for emotional aspects in care of dying”, “preparation for medical/care aspects in care of dying”, and “knowledge of supportive services and facilities” were rated as “very important” by the majority of participants. Less often rated as “very important” were “reduction of own anxiety and insecurity when dealing with the subject of death” and “contribution to the social discussion on the subject of dying”. However, all learning goals were rated as “very/rather important” by more than 60% of participants (Figure 1).

As perceived burden by death and dying might be a reason for attendance, we asked if participants felt burdened by “death and dying” at work and at home (Table 2). Summarizing it in “overall burden”, 33 of the 55 participants (60.0%) reported medium/high or very high burden by death and dying in at least one of the contexts (at home/at work).

The burden by the topic “death and dying at work” correlated with the extent of patient contact (Chi2 (df = 16) = 28.8; *p* = 0.03, Figure 2), participants reporting more frequent contact were more likely to report burden. Furthermore, we found associations of burden of participants and their learning goals: more burdened participants were likely to indicated a higher importance of “knowledge of supportive services and facilities” (Kendall tau r = −0.41; *p* = 0.00), “preparation for emotional aspects of death and dying” (r = 0.28; *p* = 0.02), and “preparation for medical/care aspects” (r = −0.25, *p* = 0.03). Thus, burdened participants reported another pattern of informational needs than participants who did not feel burdened by death and dying.

### 3.2. Evaluation of Last Aid Courses by Medical and Non-Medical Staff

The four modules of the course were mostly rated “good” to “very good” with only a small subset of participants rating the modules “not so good”, with none rating the courses being “unsatisfactory”. The best-rated module 3 “relieving suffering” was rated “very good” by 70.9% of the participants (Figure 3).

To evaluate suitability of the Last Aid course curriculum in the hospital setting, we asked the participants about their overall rating of the course: on a 4-point Likert-scale (very good, good, not so good, and unsatisfactory): 45.5% (n = 25) judged the course as “very-good”, 30.9% (n = 17) as “good”, with 23.6% (n = 13) missing answers. When the participants were asked if they would recommend the course to others, a high percentage answered positively: “applies completely” was 79.2% (n = 42), “rather applies” was 18.9% (n = 10), “does rather not apply” was 1.9% (n = 1), “does not apply” was 0%, and there were two missing answers. Likewise, most participants found that the content of the course was conveyed in an understandable manner: “applies completely” was 85.5% (n = 47), “rather applies” was 10.9% (n = 6), and there were two missing answers. Similarly, participants mostly stated that they had learned something new during the course: “applies completely” was 49.1% (n = 27), “rather applies” was 38.2% (n = 21), “rather does not apply” was 7.3% (n = 4), “does not apply” was 1.8% (n = 1), and there were two missing answers. When we correlated the answers to this question with medical competence, we found that medical staff were more likely to indicate a lower level of learning something new than non-medical staff (Kendall tau r = −0.36; *p* = 0.01).

## 4. Discussion

The Last Aid courses were well accepted and helped to reduce information deficits on care of the dying in a heterogeneous population of hospital staff, with mostly non-medical staff attending. Participants indicated emotional preparation, the need for information on supportive services, and preparation for medical and nursing aspects of end-of-life care as the most important learning goals.

We encountered gender disparities as attendees were mostly female, which is consistent with gender distribution in hospice and end-of-life care, as well as the care for individuals with dementia [10,11]. The gender disparity was also seen in a multicenter-study from Germany, Austria, and Switzerland, which conducted Last Aid courses in the community setting with more than 5000 participants [12]. This study revealed that 88% of the participants were female and the median age was 56 years, which might indicate that people attending want to prepare for a caregiver role [12]. Despite changing social expectations, females carry most of the caregiving at the end-of-life [10]. Within the context of a compassionate community, widespread Last Aid courses could—just like First-Aid courses—provide a low threshold learning opportunity that might facilitate involvement of men in end-of-life care and contribute to social change on the long run.

Interestingly, 30.9% of the Last Aid course participants had a medical background (physicians, nurses, or midwifes), although the curriculum is aimed at laypeople [12], and the courses were announced for non-medical staff on the hospital’s intranet platform. This shows the importance and public interests in the subject of death, dying, and end-of-life care. Additionally, we speculate that the interest of medical staff in Last Aid courses could indicate deficiencies in the teaching of these subjects in the training of medical professionals.

Medical (in contrast to non-medical) staff were less satisfied with the course content. When asked if “they had learned something new”, they answered with significantly lower ratings. As the Last Aid course curriculum was not designed for healthcare professionals, this finding is not surprising, but warrants an extended curriculum to meet their specific informational needs. Such a curriculum, comprising one full day with eight teaching units (each 45 min), was developed by Last Aid Germany, and is currently in the pilot-testing phase.

Furthermore, the amount of patient contact correlated with burden by death and dying. Current research gives special interest to frontline healthcare workers engaged in treating patients with COVID-19 who were at great risk of burdening symptoms of depression, anxiety, insomnia, and distress [13]; to our knowledge non-medical staff was not investigated. It is conceivable that also non-medical staff in hospitals feels burdened by death and dying especially during the COVID-19 pandemic. For medical staff, basic knowledge on palliative and hospice care is, nowadays, part of the curriculum (nurses, physicians, etc.), and advanced courses in specialized palliative care are offered in education programs. In contrast, there are no such offers for non-medical professions, even though many of them are confronted with seriously ill and dying patients (e.g., medical technologist, cleaning staff, and administration) [14]. Our results suggest that Last Aid courses are feasible to meet the informational needs in these professions. The Last Aid course curriculum addresses the core competencies recommended by the European Association for Palliative Care [15]. A recent study on online Last Aid courses for the public showed that it was feasible to hold the courses online. The results suggested that the online format enabled more people in a caregiver situation, as well as younger people, to attend [16].

Utilizing open questions, we identified and ranked reasons for participants to sign up for Last Aid courses. The three most important reasons were “preparation for emotional aspects in care of dying”, “preparation for medical/care aspects in care of dying”, and “knowledge of supportive services and facilities”. All three aspects are covered by the Last Aid course curriculum, possibly explaining why the vast majority of participants would recommend the course to others. Future studies should investigate if meeting the informational needs could contribute toward reduce the burden of death and dying in non-medical staff with patient contact. Additionally, more research on the effects of Last Aid courses on a caregiver’s willingness and capability to provide end-of-life care at home is needed.

In conclusion, Last-Aid Courses were feasible to meet the informational needs of non-medical hospital staff with high approval ratings while medical staff called for an extended curriculum.

## Figures and Tables

**Figure 1 healthcare-09-00906-f001:**
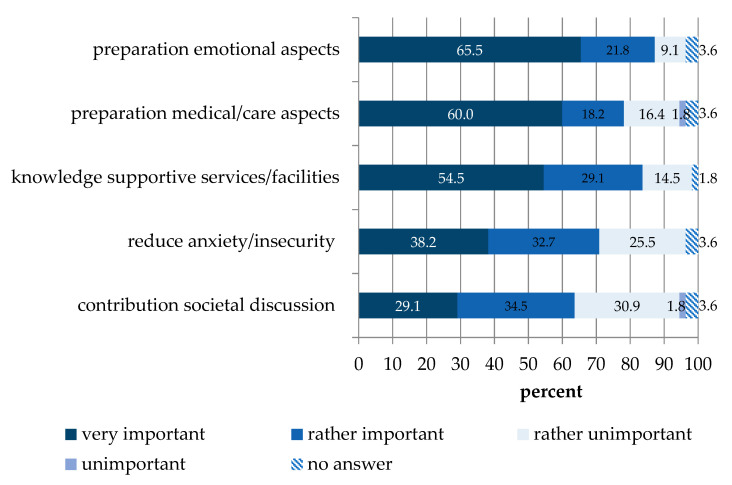
Goals and concerns before attending the course (n = 55).

**Figure 2 healthcare-09-00906-f002:**
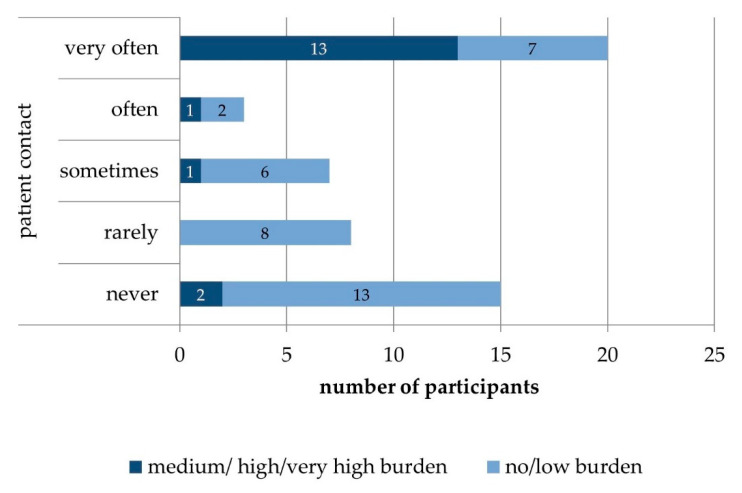
Burden by the topic death and dying at work in relation to the extent of patient contact (n = 53).

**Figure 3 healthcare-09-00906-f003:**
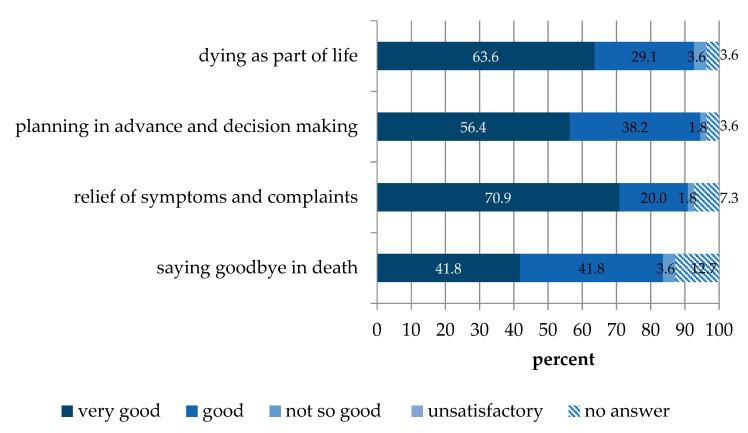
Evaluation of the Last Aid course modules (n = 55).

**Table 1 healthcare-09-00906-t001:** Characteristics of participants of the Last Aid course (n = 55).

	N (%)
**Age:**	
20–29 years	9 (16.4%)
30–39 years	4 (7.3%)
40–49 years	5 (9.1%)
50–59 years	19 (34.5%)
≥60 years	12 (21.8%)
No answer	6 (10.9)
**Sex:**	
Male	3 (5.5%)
Female	50 (90.9%)
No answer	2 (3.6)
**Medical Competence:**	
Non-medical staff	
Non-medical professions	21 (38.2%)
Therapeutic and medical assistance staff	16 (29.1%)
Medical	
Nurses and midwifes	16 (29.1%)
Physicians	1 (1.8%)
No answer	1 (1.8%)
**Contact with patients:**	
Never	17 (30.9%)
Rarely	8 (14.5%)
Sometimes	7 (12.7%)
Often	3 (5.5%)
Very often	20 (36.4%)
**Support of dying relatives:**	
Yes	20 (36.4%)
No	35 (63.6%)

**Table 2 healthcare-09-00906-t002:** Burden because of death and dying (n = 55).

	At Work N (%)	At Home N (%)
**At work**		
No burden	16 (29.1%)	13 (23.6%)
Low burden	20 (36.4%)	16 (29.1%)
Medium burden	14 (25.5%)	19 (34.5%)
High burden	2 (3.6%)	4 (7.3%)
Very high burden	1 (1.8%)	2 (3.6%)
No answer	2 (3.6%)	1 (1.8%)

## Data Availability

The data presented in this study are available on request from the corresponding author. The data are not publicly available due to privacy restrictions.

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
