# Peer review of "Lessons Learned from Introducing Last Aid Courses at a University Hospital in Germany"

_healthcare, 2021, doi:10.3390/healthcare9070906_

Round 1
Reviewer 1 Report
1) The purpose of this study (what is the relevant issue and what is the purpose to reveal it) does not seem to be clearly explained. For this reason, I found it difficult to evaluate the overall argument.
2) If Courses 1- 2 are pre-surveys to develop items for the surveys used in Courses 3-5, it would be necessary to describe in more detail the background of the 33 participants, the method of data collection, and the processes of categorization in Courses 1-2.
3) There is need to provide a reason for including medical staff in the programs for non-medical staff.
4) In the discussion section, it is written 9.4%, but what does 9.4% represent? In Table 1, Medical is written 12 (21.8%). Which is the correct number?
5) In the discussion, the author mentions the deficiencies in the training of healthcare professionals regarding death, dying and end-of-life care. However, the reasons for the medical professionals' participation in this educational course are not indicated in the body of the study. On the present description only, it would be an overstatement to mention the deficiencies in the training.
6) TableA1 suddenly appears, which is very confusing; I suggest that it be clearly marked as Appendix.
Reviewer 2 Report
TO THE AUTHORS
Thanks for submitting your work on the evaluation of the Last Aid course to Healthcare. Capacity building for staff and members of the public to support the dying person is essential to ensure a good end. I commend the authors in their attempt to find out participants’ motivation to attend and recommendations on enhancements.
The introduction is clear and the justification apparent.
I have a few suggestions that could enhance the appeal and value of the manuscript to an international readership:
Please insert line numbers in the next revision to facilitate reviewer’s work. Please do not refer to Figure numbers in the methods. That belongs in results and distracts the reader at this point. Remove the words (Figure 1) from section 2.1
There is a big gap in section 2.1 on the actual procedure of developing the survey, how the questions were built, who participated in the development, how long the process took, the occupations of the 33 participants (medical, non-medical) of people answering the open-ended questions, and whether the newly formulated evaluation questions were field tested before use.
Importantly, an Appendix with the actual questions, instruction and format (which questions were open-ended and which pre-coded) needs to be provided. This is not just out of curiosity but mainly for the reader to 1) judge whether there were the order and wording could have resulted in leading questions that could have biased the results; and 2) use the questions again in a replication study since apparently not many of the other Last Aid courses have been evaluated. The last three lines of paragraph 2.2. and the brief description in the analysis section 2.3 are not sufficient information. For instance, readers do not know if there was a definition of ‘medium, high, or very high burden by death and dying’. It is also unclear whether participants had a single understanding of ‘core/support for dying relatives’.
Results
What was the potential denominator of your target group? 60 people out of how many potential beneficiaries participated? An idea of participation rate would be informative. Why were only 5 courses offered? Not enough interest? Not enough resources? Was 100 the total numbers of staff?
Remove age category <20 years if there were no participants in it.
If the authors went through the trouble of classifying occupation in four categories (0-3), please report them as described in the analysis section.
Figure 1 is very informative, but was there an option for ‘other goal before attending the course?’.
Table 2 would be easier to read if the “at work” and “at home’ were across for ease of comparison. Table columns in this order: At work; N(%) ; At home; N(%)
Figure 3. Were the differences between not-so-good and unsatisfactory defined? If not, then merge the results into a single category
Discussion
I think the second sentence starting with “we identified goals of participants..’ is inaccurate. My understanding (without seeing the actual evaluation instrument) is that the questionnaire offered those as potential goals. It was not an open-ended question. So the claim cannot be that authors identified, but should be that participants chose goals of preparation for emotional and medical care aspects more frequently than other goals presented in the questionnaire (or something along those lines).
An explanation of why the majority of attendees were non-health professional in your study is needed; and perhaps explain in the methods precisely how the course was advertised and where.
The paragraph starting with “feeling burdened..” does not add anything new to what was already said in the results. Do not repeat the findings. Instead, please take the opportunity to compare the study with another with similar findings, such as the great example you give for the subsequent topic on Covid-19 and burden vs. patient contact. That gives great contextual meaning to your findings.
The comments on absence of courses for non-medical staff and proposal for more online approaches are a great summary of your study’s contributions to new knowledge. Well done. The suggestions for further research also sound reasonable.
It was unclear to me why you decided to invite healthcare professionals to attend if the course was not designed for them to attend in the first place. A clarification would be needed, as this might explain some of the participants’ dissatisfaction with the basic level of the information provided.
Finally, a brief overarching conclusion is missing before you close the article.
Minor: There were a few spelling mistakes and misuse of English language throughout the document which need fixing. For example ‘explorative character’ before results; ‘preperation’ in figure 1; ‘medical competence’ before Discussion, and others.
Reviewer 3 Report
The title of manuscript is: Lessons learned from introducing Last Aid Courses at a univer-sity hospital in Germany. In this study the Authors investigated the participants’ reasons for course attendance, their professional and private burden by death and dying, general feedback on the course and suggestions for improvements.
The Introduction of the study is brief and provides with some general information about the problem.
For the most part the Results section is well structured. The Discussion chapter contains information obtained after conducting experiments. The chapter is relatively synthetically described and result from the experiment. The literature used is appropriate.
